# Preparation and Application of Nano-Calcined Excavation Soil as Substitute for Cement

**DOI:** 10.3390/nano14100850

**Published:** 2024-05-13

**Authors:** Li Ling, Jindong Yang, Wanqiong Yao, Feng Xing, Hongfang Sun, Yali Li

**Affiliations:** 1Key Laboratory of Coastal Urban Resilient Infrastructures, Ministry of Education, Guangdong Provincial Key Laboratory of Durability for Marine Civil Engineering, College of Civil and Transportation Engineering, Shenzhen University, Shenzhen 518060, China; 103298691@student.swin.edu.au (L.L.); xingf@szu.edu.cn (F.X.); 2Centre for Smart Infrastructure and Digital Construction, School of Engineering, Swinburne University of Technology, Hawthorn, VIC 3122, Australia; 3Fujian Communications Planning & Design Institute Co., Ltd., Research and Development Center of Transport Industry of New Materials, Technologies Application for Highway Construction and Maintenance of Offshore Areas, Fuzhou 350000, China; yangjindong2024@gmail.com; 4State Key Laboratory of Subtropical Building Science, South China University of Technology, Guangzhou 510640, China; 202110181497@mail.scut.edu.cn

**Keywords:** excavation soil, nano-metakaolin, calcination, Portland cement, recycling

## Abstract

Rapid urbanization in many cities has produced massive amounts of problematic excavation soil. The direct disposal of untreated excavation soil often leads to significant land use and severe environmental concerns. A sustainable solution is to transform the soil waste into high-quality nano-calcined excavation soil (NCES) for application as a substitute for cement in construction. However, research in this area is very limited. This study presents a systematic investigation of the nano-sized calcined soil materials from preparation to application in cementitious material. The influence of milling parameters, including the rotational speed, milling duration, ball diameter, and milling strategy, was investigated to produce NCES with various specific surface areas. The effect of NCES substitution (15 wt% of Portland cement) in cementitious materials was then examined for mechanical performance, hydration dynamics, hydration products, and microstructure. A cement mix with very fine NCES (specific surface area of 108.76 m^2^/g) showed a 29.7% enhancement in mechanical strength and refined pore structure while a cement mix with un-grounded calcined soil showed a mechanical loss in comparison to the Control specimen. Delayed and reduced heat release at an early age was observed in a cement paste mixed with NCES. The underlying mechanism was investigated. The results of this work will contribute to the high-quality application of excavation soil waste.

## 1. Introduction

Cementitious materials are ubiquitous in modern infrastructure, where strength and durability are paramount. However, concrete structures gradually decay over time due to interconnected mechanisms including environmental exposure, mechanical damage, and intrinsic material vulnerabilities [1,2,3,4]. Therefore, nano-material admixtures emerged as a promising approach to improve the mechanical properties and durability of concrete structures [5,6,7,8].

Previously reported nano-admixtures included silica fume, ground granulated blast furnace slag, fly ash, nano metakaolin (NMK), etc. [9,10,11,12,13,14,15]. Among those nano-admixtures, NMK attracted more and more attention due to its excellent performance. For example, Morsy et al. [14] added NMK to a mortar and improved the compressive strength by approximately 8–10%. Gruber et al. [16] investigated the long-term performance of concrete containing NMK produced in North America, finding that the use of 8% and 12% NMK significantly lowered the chloride ion diffusion coefficient of concrete, on average, 50% and 60%, potentially extending the service life of reinforced concrete in chloride environments. Morsy et al. [17] reported that the addition of up to 7.5% NMK resulted in a denser matrix structure of cementitious materials, whereas a higher dosage of NMK led to the opening of the pore structure. Garg et al. [18] found that by partially replacing OPC with 7.5% NMK, not only did the compressive strength of the mortar increase by 18.2% after 28 days of curing, but its durability was also considerably enhanced due to the improved microstructure and pozzolanic activity. Zhang et al. [19] investigated the synergistic effect between NMK and OPC and found that the percentage of mesopores rose from 1% to 17% by incorporating NMK, whereas the volume fraction of large capillary pores dropped from 86% to 72%, thereby enhancing the durability of the mortar.

Currently, the NMK is mainly sourced from the mineral ore kaolinite [20,21,22]. However, the mining and transportation of kaolin ore are not only associated with high costs but also inevitably lead to a series of environmental issues [23,24]. In particular, the destruction of land resources during the mining process is particularly severe, which not only affects the local ecological balance but may also trigger chained environmental degradation [24,25]. Meanwhile, waste generation and emissions during transportation further exacerbate the pressure on environmental pollution [26]. Therefore, finding an alternative environmentally friendly and economical raw material to produce NMK has become an urgent issue for the industry to address.

The advancement of urbanization in many cities has driven massive infrastructure projects and associated excavation volumes globally [27,28]. Directly backfilling untreated excavation soil can pose environmental hazards including landslides, groundwater contamination, and greenhouse gas emissions [29,30]. Therefore, proper treatment of soil waste will benefit space release and environmental protection. Fortunately, the soil more or less has the composition of kaolinite which can transform into metakaolin through calcination [31,32]. Thus, the soil has the potential to produce nano-calcined excavation soil (NCES), which contains a certain amount of NMK, by further combining ball milling. However, it should also be noted that the soil consists of more constituents other than kaolinite, such as sand with varied particle sizes and other impurities [24,33], which might influence the preparation of NCES and its behavior in cementitious materials. Therefore, the preparation of NCES and its application in cementitious materials should be studied systematically. 

Therefore, this paper provides a systematic exploration of NCES from preparation to application with excavation soil as raw material. Firstly, the milling parameters were optimized. Then, the NCES was applied in cement as a sustainable supplementary cementitious material (SCM) to investigate its influence on the properties of cement products. Finally, the underlying mechanism was revealed. The results of this work will contribute to the high-quality application of excavation soil waste.

## 2. Materials and Methods

### 2.1. Nano-Calcined Excavation Soil (NCES) Preparation

Excavation soil was obtained from a subway construction project in Guangzhou, Guangdong Province, China (Figure 1a). After being dried at 120 °C for 2 h, the soil sample underwent crushing and sieving (150-μm) to obtain kaolinite-rich soil which was then analyzed by X-ray diffraction (XRD), thermogravimetric analysis (TGA), and X-ray fluorescence (XRF). Specifically, XRD was performed on a D8 Advance diffractometer (Bruker, Ettlingen, Germany) with a CuKα radiation source operating at a voltage of 40 kV and a current of 40 mA. Diffraction patterns were collected from 10° to 70° 2θ with a step size of 0.02° per second and a scan speed of 2°/min. TGA was conducted using a STA409PC analyzer (NETZSCH, Selb, Germany) with the soil sample being heated from 40 °C to 1000 °C at a rate of 10 °C/min under a nitrogen flow of 50 mL/min. XRF was performed with an IQ II analyzer (SPECTRO, Kleve, Germany) under vacuum conditions at voltages ranging from 25 to 50 kV and currents between 0.5 and 1.0 mA, with a dwell time of 300 s. The XRD spectrum of the sieved soil indicated that the soil consisted mainly of kaolinite and quartz (Figure 1b). The kaolinite content was determined using TGA by observing the mass loss at 400–650 °C which is associated with the conversion of kaolinite to metakaolin (Figure 1c). The XRF analysis (Table 1) showed that the content of Al and Si in the sieved excavation soil was significantly higher than that in OPC, while OPC contained more calcium. 

The sieved soil was calcined at 800 °C for 2 h followed by milling in a planetary ball mill Pulverisette 6 (Fritsch, Idar-Oberstein, Germany) with a fixed ball-to-material ratio of 10:1. Varied milling parameters, including the rotational speed (200, 350, 500 and 650 rpm), milling duration (15, 30, 45 and 60 min), zirconia milling bead diameter (0.1, 1, 5, and 10 mm), and the milling strategy, were investigated and optimized to produce NCES with specific surface areas. A schematic diagram of the NCES preparation procedure is illustrated in Figure 1d. 

The specific surface area of NCES was measured using TriStar 3000 (Micromeritics Corporation, Norcross, GA, USA). Prior to testing, the samples were ventilated at 200 °C for 4 h to thoroughly remove moisture and volatile substances. The morphology of NCES (after gold coating) was examined using a Quanta FEG 250 (FEI, Hillsboro, OR, USA) scanning electron microscope (SEM). NCES with the desired specific surface area was then applied to the cement mix as nano-metakaolin for performance evaluation.

### 2.2. Cement Paste and Mortar Preparation Using NCES

The Control cement mix in this study was prepared using ordinary Portland cement (OPC) P.I 42.5R (China United Cement Group Co., Ltd., Beijing, China), which is characterized by a minimum tricalcium silicate content of 70% and a l–e–silica ratio not less than 2.0. Its composition was determined through XRF analysis (Table 1). Sand (0.08–2 mm, from Xiamen ISO Standard Sand Co., Ltd., Xiamen, China) was used for the mortar mix. For the NCES-added cement mix, OPC was replaced by NCES as an SCM with a substitution rate of 15 wt%. The water-to-cement ratio was kept constant at 0.5 for all mixes, as well as a fixed binder-to-sand ratio of 1:3. After casting, both the paste and mortar were cured for 3, 7, 28, and 90 days at 20 ± 3 °C and relative humidity > 95% until testing.

### 2.3. Testing

The Brunauer–Emmett–Teller (BET) surface area of the prepared NCES was characterized using TriStar 3000 (Micromeritics Corporation, USA). Prior to testing, the sample was dried at 200 °C for 4 h to thoroughly remove moisture and volatile substances. The surface morphology of NCES was examined using a Quanta 250 field-emission scanning electron microscope. The instrument achieved a resolution of 1.2 nm with a high vacuum mode and an acceleration voltage of 10 kV.

The influence of NCES on the performance of the cement mortar was evaluated by mechanical (herein, compressive strength) and heat-releasing properties. The underlying mechanism was revealed through composition and porosity analysis. The compressive strength test was carried out using cement mortars with a dimension of 40 × 40 × 40 mm. The equipment used was YZH-300, which is an electronic universal testing machine manufactured by Zhejiang Luda Instruments, Shaoxing, China. During testing, the samples were loaded at a rate of 2.4 kN/s until failure occurred. The maximum load at failure was recorded and used to calculate the compressive strength. Isothermal calorimetry was used to monitor the hydration rate of cement-nano-calcined clay mixtures during early hydration, which was conducted using a TAM Air calorimeter (TA Instruments, New Castle, DE, USA). During testing, approximately 8 g of samples were loaded in glass ampoules and measured for 72 h. The heat flow and cumulative heat curves were continuously monitored.

The crystalline phase analysis of cement pastes prepared through gently grinding using an agate mortar and pestle was conducted by XRD. The functional groups of the hydration products in the cement paste were analyzed using Fourier transform infrared spectroscopy (FTIR) with a Spectrum One spectrometer (PerkinElmer, Shelton, CT, USA), which is equipped with a mid-IR source and a DTGS detector. The paste was mixed with KBr in a 1:100 ratio by mass and manually ground using an agate mortar and pestle. The mixer was then compressed into pellets with a diameter of 13 mm under a pressure of 10 tons. The transmittance spectra were collected from 4000 to 400 cm^−1^ with a resolution of 4 cm^−1^ and 32 scans. The nano-structural characterization was performed using solid-state ^29^Si magic angle spinning nuclear magnetic resonance (MAS NMR) spectroscopy on a Bruker AVANCE III 400 WB spectrometer. ^29^Si spectra were obtained using a 4 mm HR-MAS probe with ZrO_2_ rotors spun at 12 kHz. A single-pulse excitation was applied with a pulse length of 3.5 μs and an interscan recycle delay of 240 s. Chemical shifts were referenced externally to tetramethyl silane (TMS).

The pore size distribution and porosity of the cement mortars were analyzed using mercury intrusion porosimetry (MIP) on a Micromeritics AutoPore IV 9500. The applied pressure ranged from 0.1 to 30,000 psi with an equilibration time of 10 s. Additionally, the microstructure and morphology of the mortar samples prepared from a carefully broken mortar, exposing fresh surfaces, were examined using SEM. The samples were gold-coated before being loaded into the SEM chamber.

## 3. Ball-Milling Conditions for NCES Preparation

Ball-milling can effectively break NCES particles into much finer ones through the synergistic effects of impacting, shearing, and compression between the zirconia balls and the soil particles. The size of NCES powder (as characterized by a specific surface area) can be influenced by milling parameters, including the rotational speed of the grinding ball, milling time, ball diameter, and the milling strategy, etc. The effect of the aforementioned parameters was studied to obtain NCES with a powder size small enough.

The size of the NCES powder was greatly influenced by the rotation speed of the 5 mm milling balls, and an almost linear positive relationship was observed (Figure 2a). Sixty-minute milling at 650 rpm produced NCES with the highest specific surface area of 15.01 m^2^/g, which was a 53% increase from that of unmilled sieved soil of 9.82 m^2^/g (noted as N0). Based on this, 650 rpm was selected as the final rotational speed. 

The effect of milling duration on NCES’s specific area was not linear (Figure 2b). The first 15 min of milling at a rotational speed of 650 rpm increased the specific surface area of NCES by 31% from 9.82 m^2^/g to 12.90 m^2^/g. Prolonged milling until 30 and 45 min only resulted in a small increase in the specific surface area. Surprisingly, after milling for 60 min, the surface area of NCES presented a sharp growth to 15.01 m^2^/g. The milling duration was attempted to be prolonged, and it was acknowledged that after milling for 60 min at a high speed of 650 rpm, the temperature in the milling vials would rapidly increase due to prolonged frictional heating effects, which might damage the equipment. Therefore, to mitigate overheating risk to the milling equipment and to obtain NCES with possible smaller particle size, an intermittent milling procedure, i.e., 60 min milling cycles separated by 30 min cooling breaks, was adopted, as discussed in the following section.

Milling ball size (0.1, 1, 5, and 10 mm) significantly impacted NCES refinement (Figure 2c). At a fixed rotational speed of 650 rpm and milling duration of 60 min, it can be seen that when the ball diameter decreased from 10 mm to 1 mm, the specific surface area of NCES increased by 275% from 10.33 m^2^/g to 38.73 m^2^/g. The reason was that when the diameter of the milling balls decreased, more balls were required to pack the same volume. This increased the contact areas between the balls and the milled particles. The increase in contact areas promoted a more efficient transfer of energy during the ball-milling process and finally promoted a further refinement of NCES [34]. However, further decreasing the milling ball diameter to 0.1 mm resulted in NCES with a slightly lower specific surface area 32.78 m^2^/g than the 1 mm balls, which might be explained by the lack of the necessary impact force of very fine milling balls. Moreover, the smaller balls had the potential to elevate frictional forces during the milling process, thereby impeding the further refinement of the NCES powder [35]. However, the use of 0.1 mm was still worth exploring since it was proven more effective in milling finer particles due to the higher specific surface area and elevated linear velocity of the smaller balls when delivering more frequent and intense impacts, along with shear forces under identical conditions [34]. 

Intermittent milling (rotational speed of 650 rpm, with each round lasting 60 min with 30 min cooling breaks) with varied milling ball sizes was explored to produce much finer NCES (Table 2). Compared with one-round milling (Figure 3c), the two-round milling using 1 mm balls successfully increased the specific surface area of NCES by 52% to 59 m^2^/g, proving that prolonged milling would help refine the NCES particles. Interestingly, the influence of the ball diameter of either 1 mm or 0.1 mm was still insignificant (1 mm: 59.32 m^2^/g, 0.1 mm: 59.46 m^2^/g). Therefore, 1 mm balls were chosen in the 2nd round of milling for easier handling and lower cost. A 3rd round of milling using 0.1 mm balls greatly increased the surface area of NCES to 108.76 m^2^/g compared with that using 1 mm balls (69.84 m^2^/g). Consequently, a three-round intermittent milling procedure (specimen 4 in Table 2) was chosen as the optimized milling strategy to produce NCES with the smallest particle size. 

The SEM images of specimen 4 before and after each round of grinding are shown in Figure 3. It can be seen that the particle diameter of NCES reduced after each round of grinding. While the particles were at the microscale after the 1st round, they were substantially reduced to a nanoscale of approximately 200–300 nm after the 2nd and 3rd round milling.

In order to compare the influence of the particle size of NCES on the performance of cementitious materials, five cement mix designs were prepared, namely, Control, N0 N1, N2, and N3, as shown in Table 3. The Control design represented the cement mix without NCES, and N0, N1, N2, and N3 denoted the cement mix with NCES (15 wt% placement of OPC) prepared through zero rounds, one round, two rounds, and three rounds of grinding of calcined soil, respectively.

## 4. Behavior of Cementitious Materials with NCES as a Substitute

Figure 4a illustrates the compressive strength of the cement mortar incorporating NCES as substitutes. In comparison with the Control specimen, the incorporation of un-milled calcined soil decreased the compressive strength of mortars (N0) at all curing ages, although at a significantly reduced level after 28-day curing. When the calcined soil was ground into NCES, the performance of the prepared mortars was improved. Specimen N3 using the finest NCES, particularly, showed higher compressive strength than the Control specimen at all the testing ages, and a 29.7% improvement from the Control was observed in the 28-day specimen. An almost linear relationship between 28-day compressive strength and the specific surface area of NCES was observed in Figure 4b. It suggested that, in order to optimize the mechanical properties of cementitious materials, grinding the calcined soil into nanoscale was necessary. 

The measurement of hydration kinetics at early ages could be used to assess the hydration rate of cementitious materials and help evaluate the risks of cracking and damage, which is especially important for large concrete structures. The heat evolution curves within 48 h for different pastes are shown in Figure 5. Compared to the Control paste, paste with un-milled calcined soil (N0) showed delayed-reaction starting time and decreased peak height. This observation implied that the addition of calcined soil slowed down the early hydration reaction of cement, which might have been due to the dilution effect by the calcined soil with weak chemical activity at early ages [36]. It also suggested that the addition of calcined soil could help reduce the early hydration heat of cementitious materials, which is beneficial to mitigate the potential cracks caused by uneven thermal expansion stress inside and outside the structure. Milling calcined soil to nano-sized NCES and incorporating it into the cement paste (N1–3) could slightly enhance the hydration rate and heat flow, which was still much lower than that of the Control specimen. It suggests that the usage of NCES did not enhance heat release at early ages. However, it also means that finer particle size with a higher surface area accelerated the hydration reactions of OPC, possibly due to the nucleation effect [37,38].

### 4.1. Hydration Product Analysis

Phase identification of cement paste with NCES was conducted by analyzing the XRD patterns (Figure 6). From the spectra, it can be observed that with the curing age increasing from 3 days to 28 days, the peak intensity for alite in almost all specimens decreased, indicating that alite, as a primary component responsible for the early hydration of cement to form calcium silicate hydrate (C-S-H) gel and portlandite, underwent continuous consumption over time. It indicated an increase in the hydration level with time. Nonetheless, the peak intensity for alite showed no significant difference among specimens with unmilled or milled calcined soil due to the insufficient sensitivity of the XRD technique and the preferred orientation of crystalline phases. In this regard, the change in peak intensity for portlandite at 2θ values of 18.07° was assessed. Portlandite, as one of the hydration products of OPC, could be consumed by reacting with calcined soil (pozzolanic reaction) [39,40]. It was observed that the peak intensity for portlandite decreased progressively with increasing milling rounds, which indicated that the refinement of calcined soil could effectively promote the pozzolanic reaction of the system due to the increased specific surface area of NCES, which allowed it to expose more contact area with portlandite to form secondary hydration products like C-S-H and calcium aluminosilicate hydrate (C-A-S-H). Therefore, an enhancement of compressive strength was observed, as presented in Figure 6. The promotion of the hydration of OPC by NCES could also be proved by FTIR and NMR techniques. 

Results of the FTIR analysis of specimens with/without NCES are shown in Figure 7. The absorption peak observed at 3645 cm^−1^, along with a broad band of the low-intensity spread between 3000 and 3500 cm^−1^, is attributed to O-H stretching vibrations within hydrogen-bonded water molecules [41]. The peak at 1637 cm^−1^ is attributed to H-O-H bending vibration in bound water [42]. The adsorption peak appearing at 1489 cm^−1^ was related to the stretching vibrations of O-C-O due to atmospheric carbonation on the surface of the specimens [43]. The band at 1116 cm^−1^ was due to the Si-O outside of the plane-stretching vibration [44]. The weak bands at 1116 cm^−1^ corresponded to SO_4_^2−^ vibration in sulfates. The most intensive band at 970 cm^−1^ was due to Si-O-Al stretching vibration [45,46,47]. Minimal variations were observed in peak positions and types between specimens (Figure 7), implying that the incorporation of NCES had negligible influence on the chemical groups within the cement hydrates and no new phase was introduced by incorporating calcined soil or NCES. This observation could help to reduce the concern of unknown risks during the application of NCES in cementitious materials. An exception to the trend was decreased peak intensity at 3645 cm^−1^ and 3425 cm^−1^, as observed in specimens with NCES, indicating a reduction in portlandite [41] which was consistent with the finding in the XRD analysis (Figure 6).

Conventional diffraction techniques like XRD are not suitable for investigating the short-range order and nanostructure of amorphous C-S-H gels present in cementitious systems. In contrast, NMR spectroscopy provides an effective alternative approach to gain insights into C-S-H nanostructure by examining the immediate local environments of specific atomic nuclei. Magic-angle spinning (MAS) ^29^Si NMR is particularly useful as it can selectively probe the local coordination environments of Si atoms within the silicate tetrahedral units of the C-S-H gel, without being limited by the lack of long-range order. ^29^Si NMR spectra analysis could offer insights into the connectivity of silicate chains (denoted by Q^i^ tetrahedra) and substitution mechanisms. The resonance at approximately −71.2 ppm (Q^0^) signifies an isolated silicon–oxygen tetrahedron, suggesting a state of partial reaction or the hydrolysis of silicates [48]. The distinct peaks located around −79.5 ppm (Q^1^) and −83.5 ppm (Q^2^) represent Si tetrahedral units linked with one and two other Si tetrahedra species, respectively [49].

The ^29^Si MAS NMR spectra of the 28-day cement specimen were recorded and presented in Figure 8. The deconvolution of the spectra highlighted that the silicate units in the C-S-H gel formed during cement hydration predominantly existed as Q^0^, Q^1,^ and Q^2^ configurations [48]. In the cement mix with calcined soil (N0), the intensity of the Q^0^ peak markedly decreased, implying the enhanced dissolution of the unhydrated silicate phases (C_3_S, C_2_S) and the concurrent formation of C-S-H gel as the hydration product. Furthermore, a distinct shoulder, centered around −82 ppm (assigned as Q^2^(1Al)), appeared between the Q^1^ and Q^2^ sites. The presence of Q^2^(1Al) units indicated the partial substitution of Al^3+^ for Si^4+^ within the C-S-H gel structure, demonstrating the formation of C-A-S-H gels in the cement hydration products. With the addition of the NCES of the increased specific surface area in the cement mix, the intensity of this Q^2^(1Al) peak significantly increased with a concurrent reduction in Q^1^ peak intensity, which signified enhanced Al substitution within C-S-H from the reactive aluminosilicate phases in NCES. This MAS NMR analysis demonstrated the formation of Al-substituted C-A-S-H gels.

Deconvolution of the ^29^Si NMR spectra quantifies the proportions of the silicate species Q^0^, Q^1^, Q^2^, and Q^2^(1Al) in various 28-day cement mixes, as presented in Table 4. Table 4 also presents the degree of hydration (α), mean silicate chain length (Ψ), and Al/Si substitution ratio calculated using Equations (1), (2), and (3), respectively, as fundamental parameters for comprehensively elucidating the hydration mechanism and C-A-S-H microstructural evolution [50]. N0 (cement specimen with unmilled calcined soil) showed decreased Q^0^ resonance and dramatically augmented Q^2^(1Al) resonance compared to the Control specimen (Table 4). Further elevation of the calcined soil surface area (as prepared NCES) consistently augmented the proportion of Q^2^(1Al) sites and increased Ψ, Al/Si, and α values. The enhanced hydration may be attributed to the high reactivity and solubility of the fine NCES particles releasing aluminum and silicon species into the pore solution to interact with the cement phases [51]. The increased availability of dissolved ions raised the solution supersaturation, driving faster precipitation of the C-A-S-H gel products. The increased Ψ and Al/Si ratios signified the elongation of the mean C-A-S-H chain length over the 28-day hydration period. Such observations were consistent with established ^29^Si NMR principles, where specific [SiO_4_]^4−^ configurations exhibited distinct chemical shifts and peak intensities.

The results implied that the NCES actively participated in the hydration reactions and substantially modified the structural characteristics of the C-A-S-H gels formed within the cement–metakaolin blends. The increased solubility and surface area enabled the NCES to accelerate hydration while also promoting the formation of C-A-S-H binder phases with extended silicate chain polymerization and heightened aluminum incorporation. These microstructural refinements, realized by incorporating NCES with a higher surface area, can potentiate significant improvements in the mechanical performance of the cementitious systems [52].
(1)$$\Psi =\frac{I(Q^{1})+I(Q^{2})+\frac{3}{2}I[Q^{2}\left(1Al\right)]}{\frac{1}{2}I(Q^{1})}$$
(2)$$Al∕Si=\frac{0.5\times I[Q^{2}\left(1Al\right)]}{I(Q^{1})+I(Q^{2})+I[Q^{2}\left(1Al\right)]}$$
(3)$$\alpha =\left[100-I\left(Q^{0}\right)\right]\times 100\%$$

### 4.2. Microstructural Analysis

The cumulative pore size distribution obtained through the MIP of the cementitious materials of specimens with/without NCES was studied and the results were shown in Figure 9. The Control specimen showed a pore size distribution concentrated between 500–5000 nm, while the addition of un-grounded calcined soil (N0 specimen) led to higher total porosity but a left-shift of pore size distribution and average pore sizes between 100–3000 nm. The specimens with NCES showed a reduction in both the total porosity and average pore size compared to the Control and N0. Specimen N1, for example, had a similar total porosity to the Control but a markedly lower average pore size centered around 100–1000 nm. Specimen N2 and N3 had similar pore size ranges between 80–300 nm, but N3 showed notably lower overall porosity than N2. The porosities of N3 and N2 were reduced by 38.75% and 20.86% compared with the Control specimen, respectively. The reduction in porosity and finer pore structures for specimens with NCES can be attributed to the higher pozzolanic reactivity and filler effect resulting from the higher specific surface area. The densification of the pore structure was usually beneficial for mechanical performance improvement, as evidenced by the enhanced compressive strength of specimens with NCES (Figure 6).

SEM was utilized to further investigate the influence of the NCES substitute on the hydration products and microstructural of the cement pastes after 28 days of curing. Representative SEM micrographs of the Control specimen and NCES-blended cement pastes are presented in Figure 10. The Control sample exhibited typical hydration products, as expected in OPC pastes, namely, C-S-H as a dense semi-amorphous matrix binding the hydrated particles and crystalline portlandite (CH) as distinct hexagonal platelets with well-defined smooth edges, indicating regulated hexagonal crystal growth during the hydration process. Minor pores were visible within the microstructure. In contrast, significant microstructural differences were observed in the nano-calcined soil-blended cement pastes. Firstly, the microstructure appeared denser and more compact compared to the Control specimen. This enhanced density was attributed to the pozzolanic reaction between the reactive metakaolin component in NCES and CH, resulting in an additional C-S-H formation within the available pore space. Secondly, the characteristic hexagonal CH crystals prominent in the Control specimen were significantly reduced. The remaining CH crystals exhibited rounded edges and shrinking volumes, indicating destabilization and dissolution. This confirmed the pozzolanic consumption of CH by the metakaolin. Additionally, sparse formation of the C-A-S-H phase was observed at the boundaries of partially dissolved CH crystals. The C-A-S-H was indicative of the pozzolanic reaction between NCES and CH. The altered CH morphology and presence of C-A-S-H signified the progression of pozzolanic reactions and the formation of additional strength-enhancing hydration products. Overall, the SEM analysis demonstrated the pronounced effect of NCES in refining the microstructure, consuming CH, and generating additional hydration products through pozzolanic reactions. The microstructural difference correlated well with the improved mechanical performance observed in the NCES-blended specimens, as shown in Figure 6.

## 5. Discussion

Regarding mechanical properties, before being ground to nano-sized particles, the calcined excavation soil replaced 15% of OPC. This substitution resulted in a reduction in the compressive strength of the blended specimens at all tested ages, from 3 days to 28 days. However, when the calcined soil was ground into nano-size, the compressive strength of the blended specimen increased, and the increasing level was highly dependent on the particle size of NCES. The blended specimen with the finest NCES (N3) even showed a compressive strength higher than the Control specimen without the calcined soil. 

The excellent mechanical behavior of the NCES-blended specimen resulted from two aspects. One was the Control of the pore structure. When the calcined soil was ground to nano-size, it presented much higher surface areas, which provided more reactive sites for either the hydration of OPC or the pozzolanic reactions between the NCES and CH released during cement hydration, accelerating the formation of C-S-H gel and/or C-A-S-H gels to fill the pores. Moreover, the nano-sized NCES particles could also fill the capillary pores down to nano-size. Both densified the microstructure of NCES-blended specimens and improved the mechanical properties. The other aspect is the modification of hydration products. The refinement of NCES enabled the NCES-blended specimen to transform partial C-S-H to a denser C-A-S-H and increase the mean C-A-S-H chain length. Both also densified the matrix of NCES-blended specimens and improved their mechanical property.

Therefore, the ground of the calcined soil through ball milling is significantly beneficial to the improvement of the mechanical properties of NCES-blended specimens through pore structure and composition modification. Although the early heat released increased a little bit, it was still much lower than that of the Control specimen.

## 6. Conclusions

This paper presents a systematic investigation of the nano-sized calcined soil materials from preparation to application. Based on the results, the main conclusions are presented as follows.

During the preparation of NCES, the size of NCES powders was found to be significantly influenced by milling parameters, including the rotational speed of the grinding ball, milling time, ball diameter, and the milling strategy. Finally, the milling parameters were optimized, and a three-round milling strategy was adopted to prepare NCES with the finest particle size and highest specific surface area of 108.76 m^2^/g.

After the NCES was loaded into cementitious materials as a partial replacement (15%) of OPC, the mechanical and heat-releasing properties were found significantly affected by the fineness of NCES. The grinding of NCES could effectively compensate for the mechanical loss caused by the calcined soil while still keeping a much lower heat release at an early age. It suggested that the full refinement of the calcined soil to nano-size was beneficial for the high-quality application of the excavation soil waste.

Through investigating the underlying mechanism, it was found that the improvement of the performance of NCES-blended cementitious materials was closely related to pore structure and composition modification. The NCES helped densify the micro-structure of cementitious materials through an occurring pozzolanic reaction between NCES and CH, promoting the hydration of OPC, and filling the pores between hydration products. Meanwhile, the transformation of C-S-H to a denser C-A-S-H and the increase in the mean C-A-S-H chain length were also helpful for the strength enhancement of the cementitious materials. Here, it should be noted that the improvement in pore structure also improved the durability of the concrete structure.

Generally, this study proved the high-quality application of excavation soil waste through producing NCES as SCM to enhance the 28-day compressive strength and microstructure of cementitious materials. Future investigation should focus on understanding the property change of the impurities in calcined soil during intermittent milling and their influence on cement product performance. Additionally, the long-term strength and durability of cementitious materials using NCES should be investigated with attention to identifying/understanding the positive/negative influence of the impurities. Furthermore, the preparation conditions of NCES should be optimized for economic and environmental benefits. 

## Figures and Tables

**Figure 1 nanomaterials-14-00850-f001:**
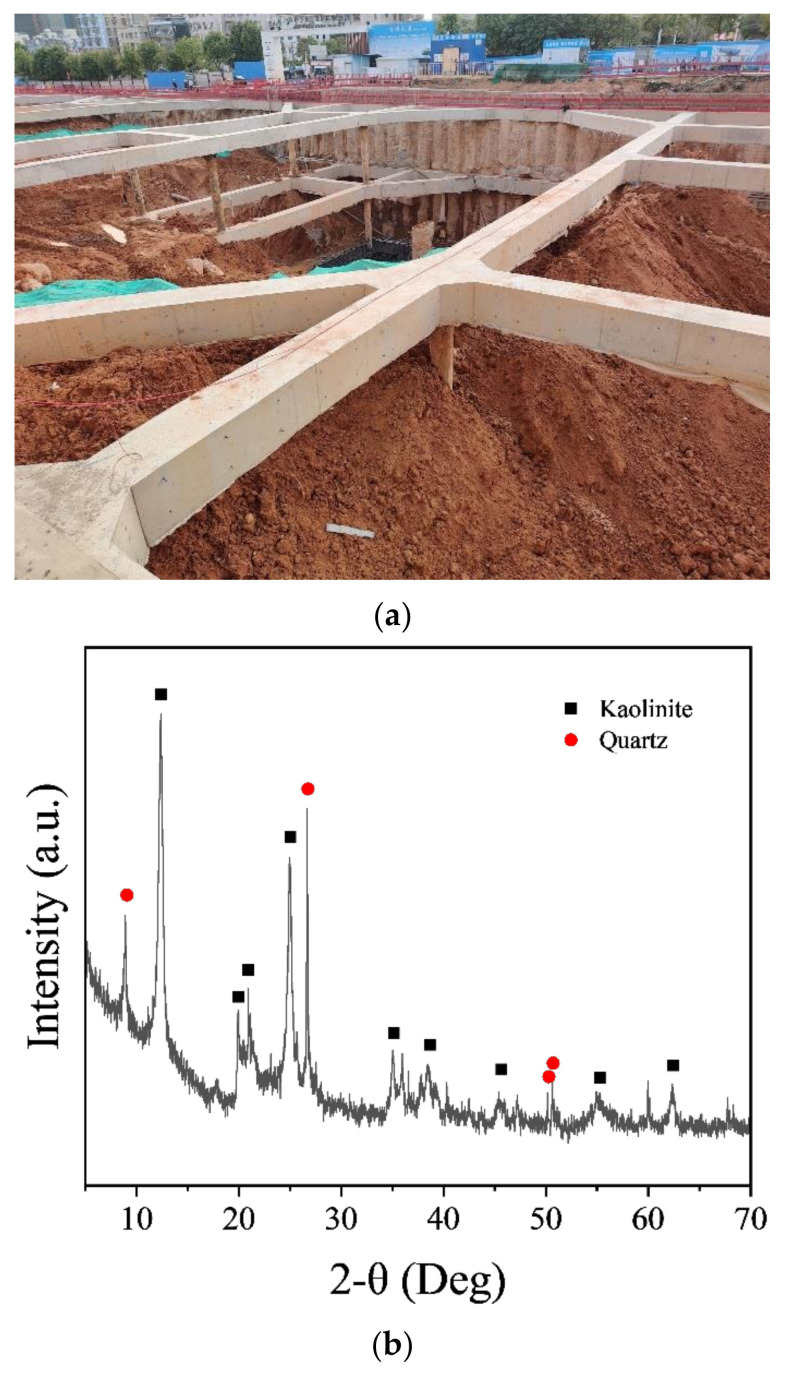
Collection site (**a**), XRD (**b**), and TGA (**c**) spectrum of sieved excavation soil, and (**d**) the preparation procedure of the NCES suspension.

**Figure 2 nanomaterials-14-00850-f002:**
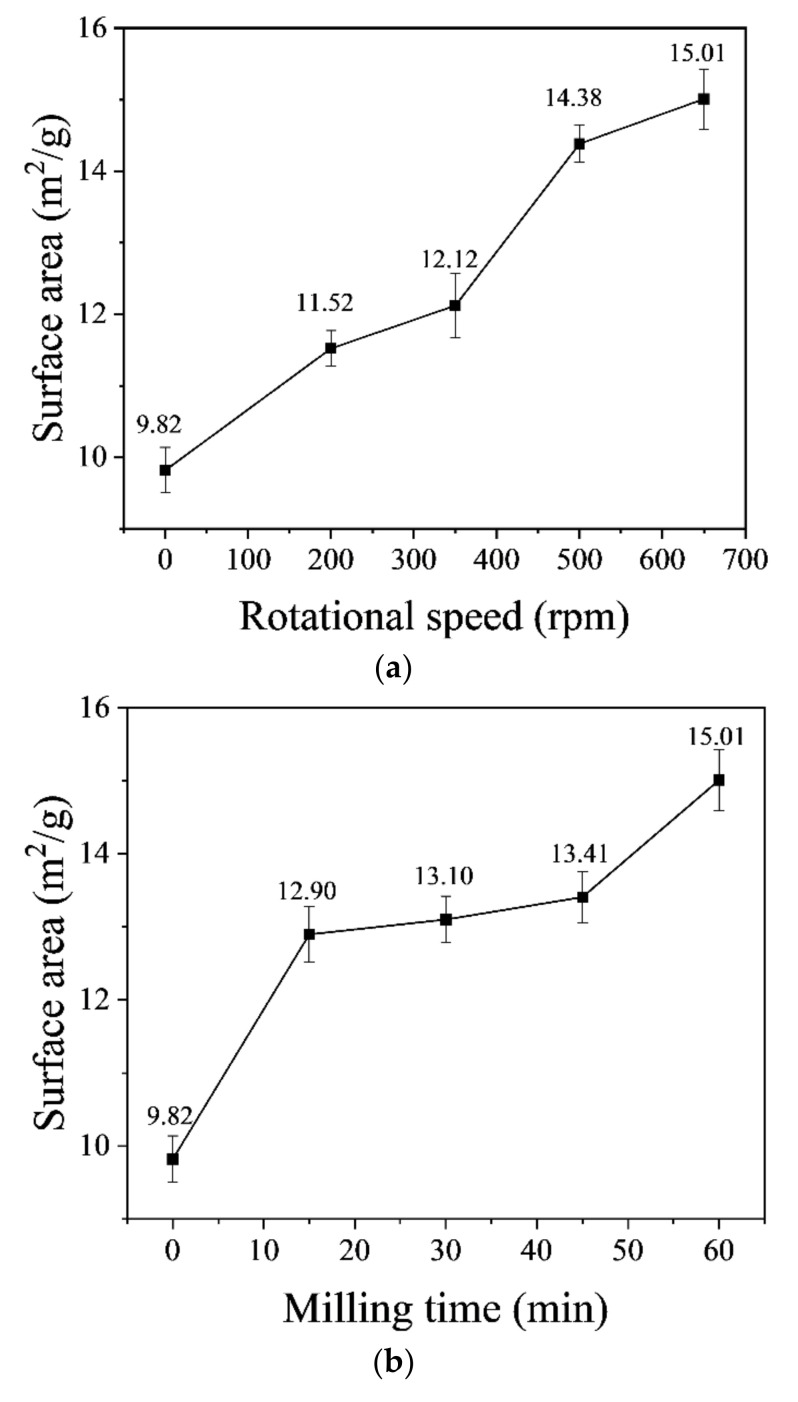
The specific surface area of NCES obtained at (**a**) varied rotational speed (milling duration 60 min, ball diameter 5 mm), (**b**) varied milling time (rotational speed 650 rpm, milling ball diameter 5 mm), and (**c**) varied milling ball diameter (rotational speed 650 rpm, milling duration 60 min).

**Figure 3 nanomaterials-14-00850-f003:**
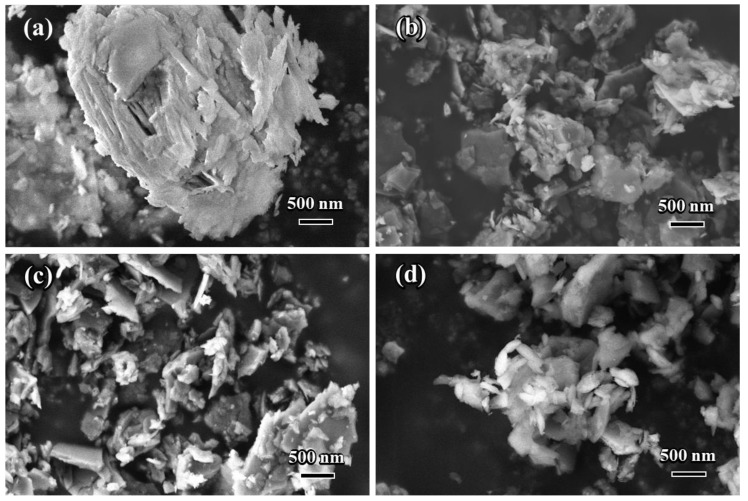
SEM images of calcined excavation soil before grinding (**a**) and the prepared NCES after the 1st round (**b**), 2nd round (**c**), and 3rd round (**d**) of grinding.

**Figure 4 nanomaterials-14-00850-f004:**
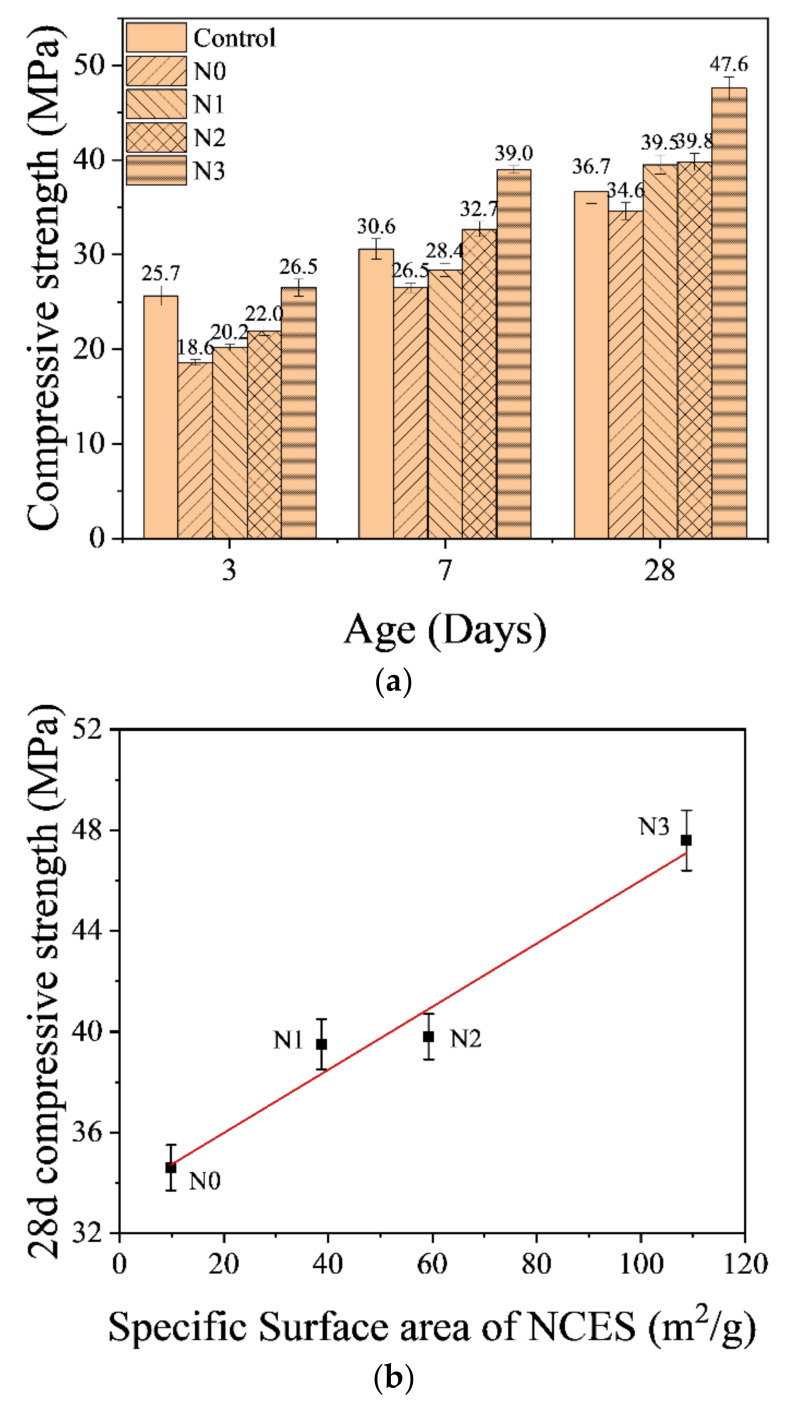
(**a**) Compressive strength of specimens with NCES as a substitute. (**b**) Relationship between compressive strength of 28-day mortar and surface area of NCES.

**Figure 5 nanomaterials-14-00850-f005:**
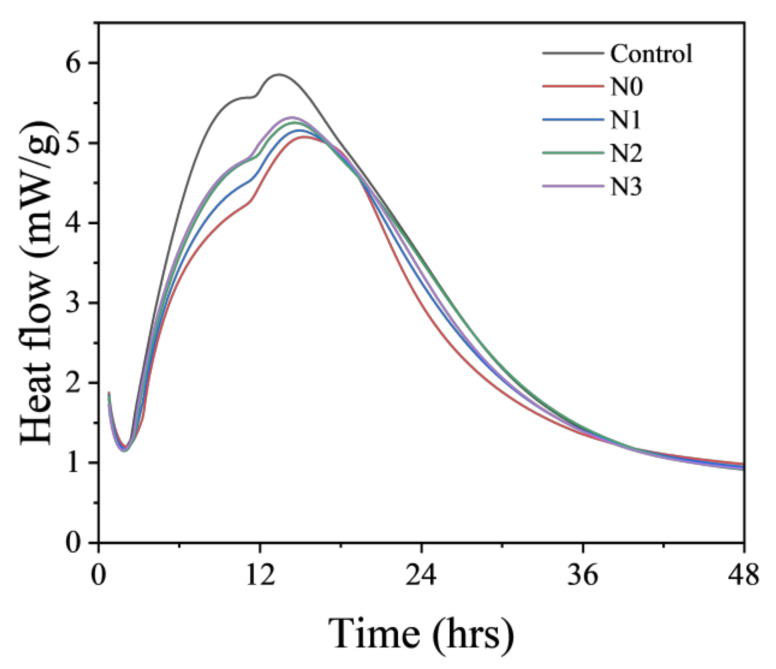
Heat flow of specimens with NCES as a substitute.

**Figure 6 nanomaterials-14-00850-f006:**
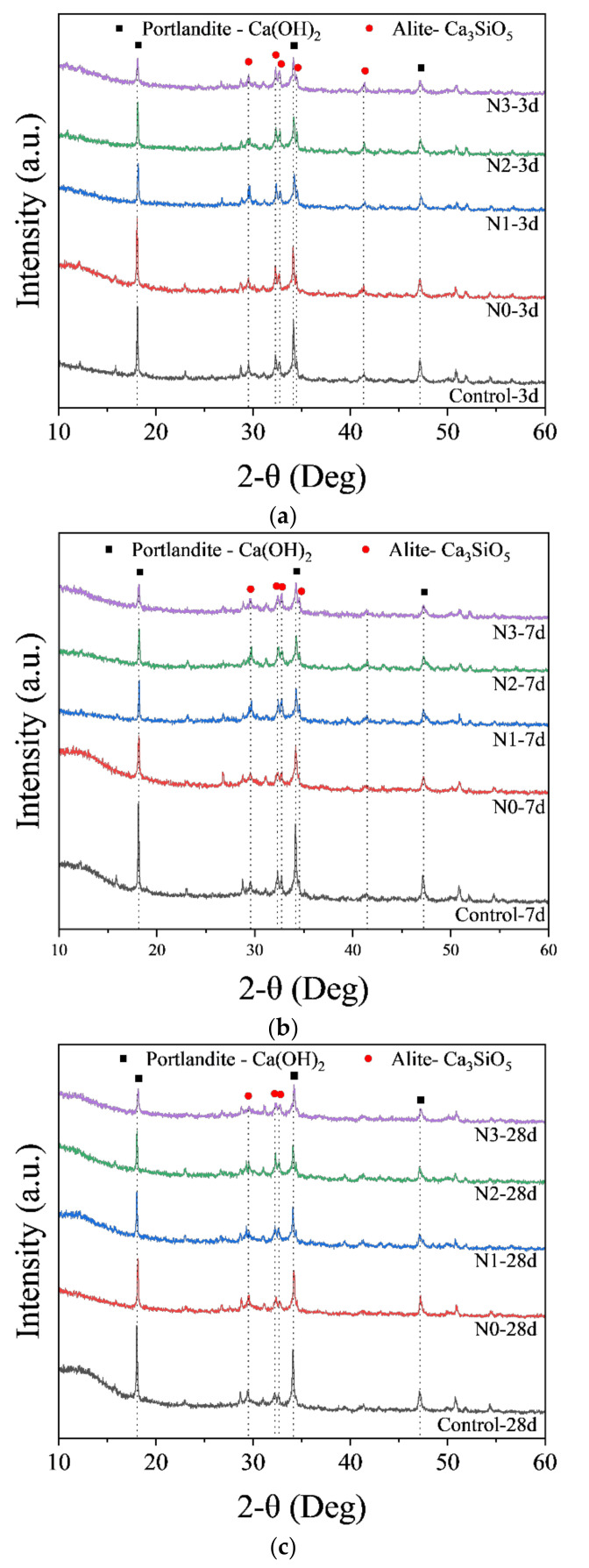
XRD spectra of specimens with/without NCES as a substitute: (**a**) 3d; (**b**) 7d; and (**c**) 28d.

**Figure 7 nanomaterials-14-00850-f007:**
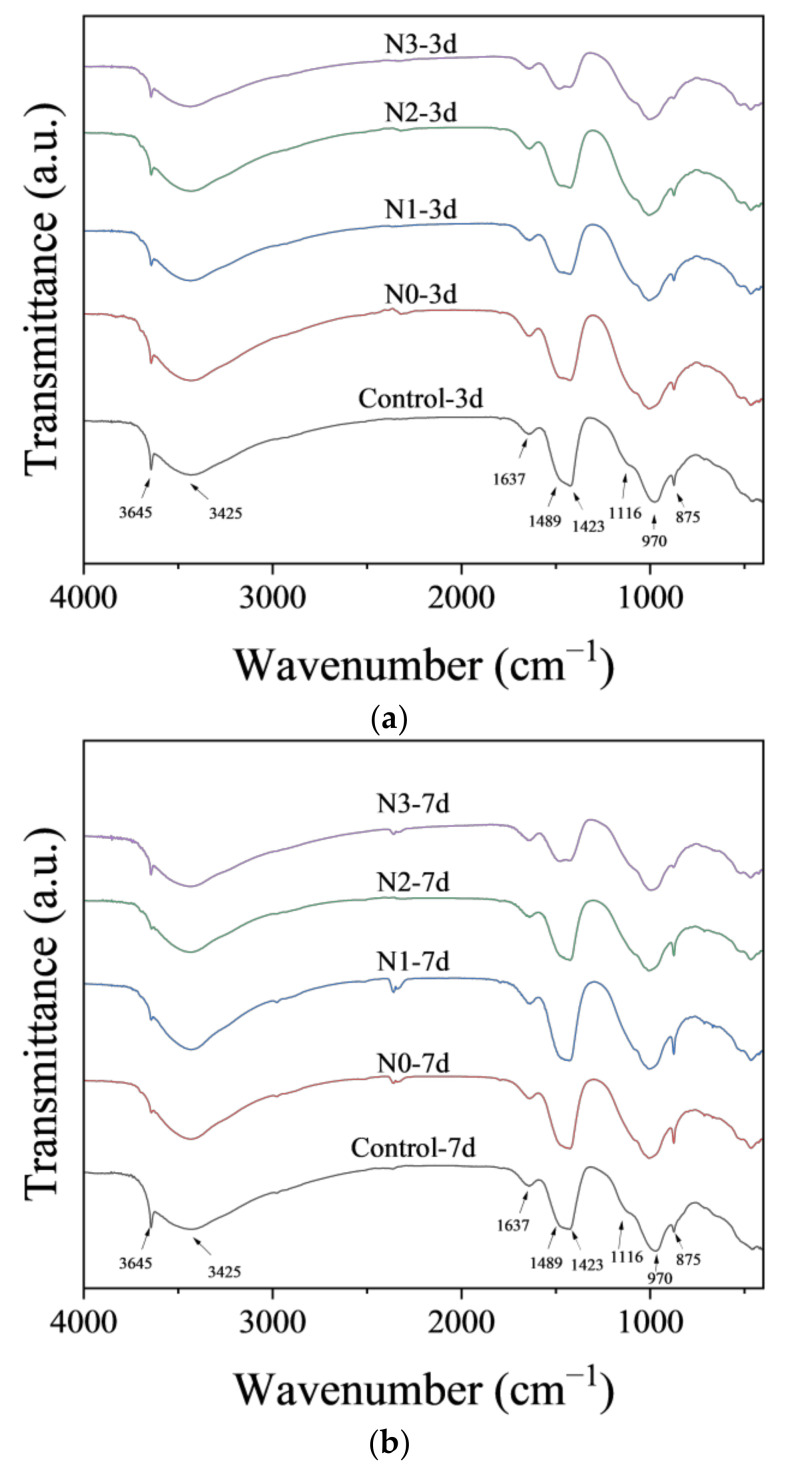
FTIR spectra of specimens with/without NCES as a substitute: (**a**) 3d; (**b**) 7d; and (**c**) 28d.

**Figure 8 nanomaterials-14-00850-f008:**
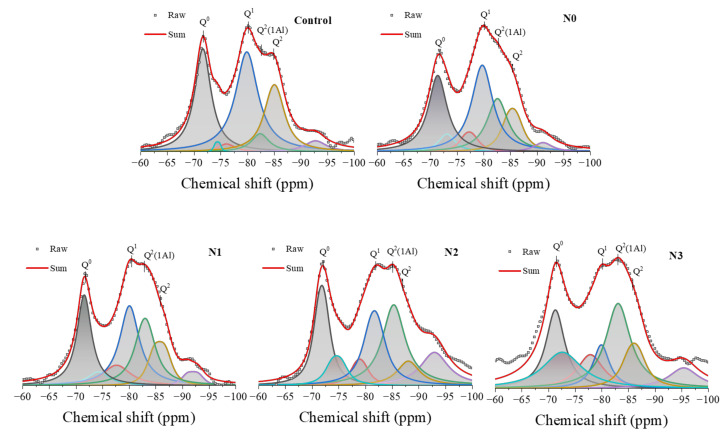
Si NMR spectra along with deconvolutions and fitting for cement specimens at 28-day curing age. The original spectrum is shown in red. Deconvoluted peaks are highlighted in different colors for clarity: Blue represents Q^0^, yellow denotes Q^1^, cyan indicates Q^2^ (1Al), and brown marks the remaining Q^2^ peaks.

**Figure 9 nanomaterials-14-00850-f009:**
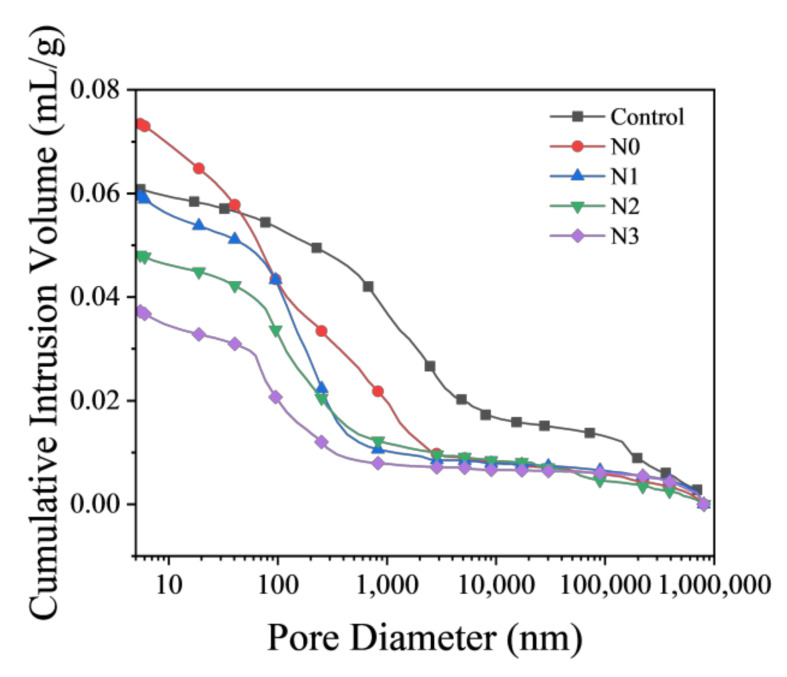
Pore size distribution of specimens.

**Figure 10 nanomaterials-14-00850-f010:**
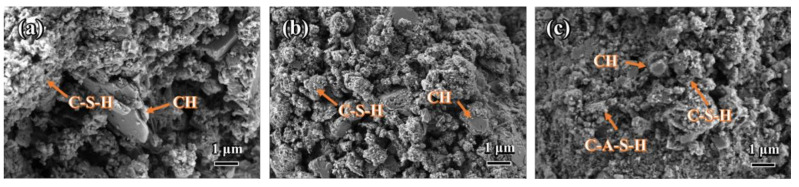
SEM images of the Control (**a**), N0 (**b**), and N3 (**c**) mortar specimens after 28 days of curing.

**Table 1 nanomaterials-14-00850-t001:** Chemical composition of the excavation soil and OPC in oxide form (from XRF analysis).

Chemical Composition (wt%)	SiO_2_	Al_2_O_3_	Fe_2_O_3_	K_2_O	CaO	TiO_2_	MgO	SO_3_	Na_2_O
Sieved excavation soil	50.26	43.91	4.25	0.92	0.02	0.33	0.07	0.10	-
OPC	21.77	4.62	3.62	-	64.68	-	2.80	0.46	0.50

**Table 2 nanomaterials-14-00850-t002:** The surface area of NCES after intermittent milling at a rotational speed of 650 rpm.

Specimen No.	Milling Ball Diameter (mm)			Surface Area (m^2^/g)
	1st Round 60 min	2nd Round 60 min	3rd Round 60 min	
1	1	1	-	59.32
2	1	0.1	-	59.46
3	1	1	1	69.84
4	1	1	0.1	108.76

**Table 3 nanomaterials-14-00850-t003:** Cement mix designs with various NCES.

Cement Mix Design	NCES (wt% of OPC)	NCES Surface Area (m^2^/g)	Milling Ball Size (mm) *		
			Round 1 60 min	Round 2 60 min	Round 3 60 min
Control	0	-	-	-	-
N0	15	9.87	-	-	-
N1	15	38.73	1	-	-
N2	15	59.32	1	1	-
N3	15	108.76	1	1	0.1

Note: * Rotational speed remained at 650 rpm.

**Table 4 nanomaterials-14-00850-t004:** Analytical results of the deconvolutions of ^29^Si NMR.

Specimen	Surface Area (m^2^/g)	The Cumulative Integrated Intensity/%				Ψ	Al/Si	α/%
		I (Q^0^)	I (Q^1^)	I (Q^2^)	I (Q^2^(1Al))			
Control	-	28.28	37.63	21.51	5.43	3.58	0.04	71.72
N0	9.87	24.35	30.83	12.81	18.88	4.67	0.15	75.65
N1	38.73	21.99	26.30	11.93	21.82	5.40	0.18	78.01
N2	59.32	19.92	20.52	7.80	25.99	6.56	0.24	80.08
N3	108.76	18.22	8.82	11.05	25.85	13.29	0.28	81.78

Note: I(Q^0^), I(Q^1^), I(Q^2^), and I(Q^2^(1Al)): the cumulative integrated intensity for peak Q^0^, Q^1^, Q^2^, and Q^2^(1Al), respectively; Ψ: mean silicate chain length; Al/Si: aluminum-to-silicon ratio; α: degree of hydration.

## Data Availability

Data are contained within the article.

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
