# Peer review of "Preparation and Application of Nano-Calcined Excavation Soil as Substitute for Cement"

_nanomaterials, 2024, doi:10.3390/nano14100850_

Round 1
Reviewer 1 Report
Comments and Suggestions for Authors
Although the topic of this article is interesting and significant to the scientific community, but I would like to take minor review from the authors before acceptance. The comments on this article that is to be answered are:
1. Captions should be made for Figure 3 (a, b, c, d), and not written on the figures themselves.The same applies to Figure 10.
2. Section 2.1 describes the X-ray spectrum (Fig. 1b) and thermogram (1c). However, the description of instruments and techniques comes much later (section 2.3). It is not right. First, it is necessary to describe the instruments and research methods, and then describe the results.
3. It is necessary to explain how the FTIR spectra were obtained. What did the authors use for analysis—powder as for XRF or the samples of cement composites themselves?
4. In Figure 7, the y-axis has units of %, but there are no values. It is necessary to correct the units of measurement to a.u.
5. It is not entirely clear why the authors chose a grinding time of 60 minutes. Indeed, in Figure 2 b, it is noticeable that the specific surface area increases with increasing grinding time and at 60 minutes the curve does not reach a plateau. Perhaps grinding for 2 hours showed even better results?
Reviewer 2 Report
Comments and Suggestions for Authors
This manuscript presented a systematic investigation of the nano-sized 21 calcined soil materials from preparation to application in cementitious material. Some issues should be addressed.
1. In table 2, how did the authors determine the SA?
2. What is the resolution in Fig. 3?
3. How did the author determine the heat flow? Besides, why the five curves converted to the same values at 48 hours?
4. What's the meaning of numbers (i.e., 1637, 970) in Fig. 7. In addition, The ordinate value of (transmittance) is missing.
5. What are the meaning of symbols in Eqs. (1-3)?
6. Introduction part. Some relevant investigations should be added to make a more complete literature review, e.g., Construction and Building Materials (2022), 320, 126211.
7. In the last section, the author mainly summarized some valuable conclusions. What’s the critical significance and limitation of this work?
8. The language used in this manuscript also needs to be thoroughly revised (such as the capital and small letter, unit, …).
Comments on the Quality of English LanguageThe language used in this manuscript also needs to be thoroughly revised (such as the capital and small letter, unit, …).
Round 2
Reviewer 2 Report
Comments and Suggestions for Authors
The manuscript is significantly improved after revision, and it can be accepted now.